# ERK-Smurf1-RhoA signaling is critical for TGFβ-drived EMT and tumor metastasis

Jianzhong Zheng[1,2,*] , Zhiyuan Shi[1,2,*], Pengbo Yang[6], Yue Zhao[1,2], Wenbin Tang[1,2] , Shaopei Ye[1,2], Zuodong Xuan[1,2], Chen Chen[1,2], Chen Shao[2], Qingang Wu[1,3] , Huimin Sun[4,5]

**Epithelial-mesenchymal transition (EMT) has fundamental roles in various biological processes. However, there are still questions pending in this fast-moving field. Here we report that in TGFβ-induced EMT, ERK-mediated Smurf1 phosphorylation is a prerequisite step for RhoA degradation and the consequent mesenchymal state achievement. Upon TGFβ treatment, activated ERK phosphorylates Thr223 of Smurf1, a member of HECT family E3 ligase, to promote Smurf1-mediated polyubiquitination and degradation of RhoA, thereby leading to cell skeleton rearrangement and EMT. Blockade of phosphorylation of Smurf1 inhibits TGFβ-induced EMT, and accordingly, dramatically blocks lung metastasis of murine breast cancer in mice. Hence, our study reveals an unknown role of ERK in TGFβ-induced EMT and points out a potential strategy in therapeutic intervention.**

## Introduction

Epithelial–mesenchymal transition (EMT), which plays pivotal roles in embryonic development, wound healing, organ fibrosis and even cancer progression (Nieto et al, 2016; Chen et al, 2017), is a very complicated program whereby epithelial cells loss cell–cell contacts, having mesenchymal characteristics and dissociating from their original sites (Kalluri & Weinberg, 2009; Lamouille et al, 2014). Furthermore, tumor cells acquire cancer stem cell and get the properties of chemo resistance through EMT (Mani et al, 2008; Fischer et al, 2015; Zheng et al, 2015). EMT could be induced transcriptionally and post-translationally by different kinds of grow factors including EGF, VEGF, hepatocyte growth factor (HGF), FGF, insulin-like growth factor (IGF), PDGF, and transforming growth factor (Yang et al, 2006; Xu et al, 2009; Chung et al, 2011; Smith et al, 2011; Al Moustafa et al, 2012; Wu et al, 2013; Katoh & Nakagama, 2014; Li et al, 2017). Among all these kinds of signaling pathways, TGFβ signaling is the most important and well-characterized signaling cascade. TGFβ signaling includes smad-dependent and smad-independent pathways, and nearly it has remarkable effects in the regulation of epithelial transdifferentiation process in all the scenarios in which EMT happens (Derynck & Zhang, 2003; Derynck et al, 2014; Gonzalez & Medici, 2014; Chaikuad & Bullock, 2016; Hata & Chen, 2016).

MAPK signal pathways, including ERK, p38, and JNK MAPKs in mammalian cells, activated through MAPKKK, MAPKK, and finally MAPK, convert multiple extracellular stimuli into intracellular cascades and biological outcomes (Johnson & Lapadat, 2002). ERK MAPK could be activated by diverse growth factors (e.g., EGF, HGF, IGF, PDGF, and TGFβ) and regulated at its origin by Ras GTPases, which lead to the activation of MAPKKK, constituted by Raf family kinases. Activated MAPKKK further phosphorylate and activate the dual-specific kinases MEK1 and MEK2, and these two kinases in turn activate ERK1 and ERK2 by phosphorylation (Grimberg & Cohen, 2000; Pinzani, 2002; Raman et al, 2007; Derynck et al, 2014; Pachmayr et al, 2017; Guo et al, 2020). ERK has important roles in cell proliferation, differentiation, stress response and apoptosis by phosphorylating its substrates in the nucleus or in the cytoplasm (Yoon & Seger, 2006). Numerous studies point out that ERK pathway plays notable roles in cancer progression and tumor metastasis by its nuclear functions to regulate gene expression (Plotnikov et al, 2011, 2015; Maik-Rachline et al, 2019). However, whether ERK has any non-nuclear functions in this pathological process still remains largely unknown.

In this study, we demonstrate that in response to TGFβ treatment, activated Erk1/2 phosphorylates E3 ligase Smurf1, thereby promoting its binding to RhoA and subsequent ubiquitination and degradation, which is critical for cell–cell junction dissociation. Our study uncovered a new mechanism underlying TGFβ-induced EMT, providing a new insight for fully understanding the regulation of epithelial cell plasticity during EMT.

[1]School of Medicine, Xiamen University, Xiamen, China   [2]Department of Urology, Xiang'an Hospital of Xiamen University, School of Medicine, Xiamen University, Xiamen, China   [3]Zhejiang Provincial Key Laboratory of Pancreatic Disease, The First Affiliated Hospital, and Institute of Translational Medicine, Zhejiang University School of Medicine, Hangzhou, China   [4]The Central Lab of Xiang'an Hospital of Xiamen University, School of Medicine, Xiamen University, Xiamen, China   [5]The Key Laboratory for Endocrine Related Cancer Precision Medicine Of Xiamen, Xiang'an Hospital of Xiamen University, School of Medicine, Xiamen University, Xiamen, China   [6]Key Laboratory of Birth Defects and Related Diseases of Women and Children of MOE, State Key Laboratory of Biotherapy, West China Second University Hospital, Sichuan University, Chengdu, China

Correspondence: hmsun@xah.xmu.edu.cn; Qingangwu@126.com
*Jianzhong Zheng and Zhiyuan Shi contributed equally to this work.

# Results

### ERK interacts with Smurf1

To deeply find out the biological functions of ERK and the underlying mechanisms, we first carried out affinity purification of Flag-ERK2 using MCF-7 cells, followed by mass spectrometry analysis to identify the new binding partners of ERK2. Among the interacting proteins, we discovered the HECT family E3 ligase Smurf1 (Fig 1A). To verify the interaction, we used Flag-Smurf1 and Flag-Smurf2 to examine their interactions with HA-ERK1 or HA-ERK2 by co-ip assay. We found that both ERK1 and ERK2 could specifically bind to Smurf1 but not Smurf2 (Figs 1B and S1A). We further confirmed that Flag-ERK2 coprecipitated with endogenous Smurf1 (Fig S1B). Moreover, we used endogenous ERK antibody to immunoprecipitate endogenous ERK from MCF7 cell lysate, and ascertained that Smurf1 indeed interacted with ERK endogenously (Fig 1C).

Next, we performed GST pull-down assay to examine the interaction between ERK and Smurf1 in vitro. As predicted, bacterially purified

GST-ERK2 could directly interact with His-Smurf1 (Fig 1D). Furthermore, we detected that ERK and Smurf1 could colocalize with each other in the cell membrane by using immunofluorescence assay (Fig 1E).

### ERK mediates the phosphorylation of Smurf1 on Threonine 223

Previous studies showed that Smurf1 usually targets its substrates for polyubiquitination and degradation (Wang et al, 2003, 2006; Cao & Zhang, 2013; Fu et al, 2020; Xia et al, 2021). Thus, we wanted to know whether Smurf1 influences ERK protein levels. Overexpression of Smurf1 did not affect the steady-state levels of both ERK1 and ERK2 (Fig S2A and B). Instead, we observed that Smurf1 was phosphorylated in response to TGFβ treatment, and this could be blocked by MEK inhibitor U0126 (Fig 2A and B), indicating ERK plays a role in regulating Smurf1 phosphorylation in this process. In good line with this, TGFβ treatment notably enhanced the interaction between Smurf1 and ERK (Fig S2C). We performed in vitro kinase assay using both constitutively active form of ERK2 (ERK2-R67S) and catalytically

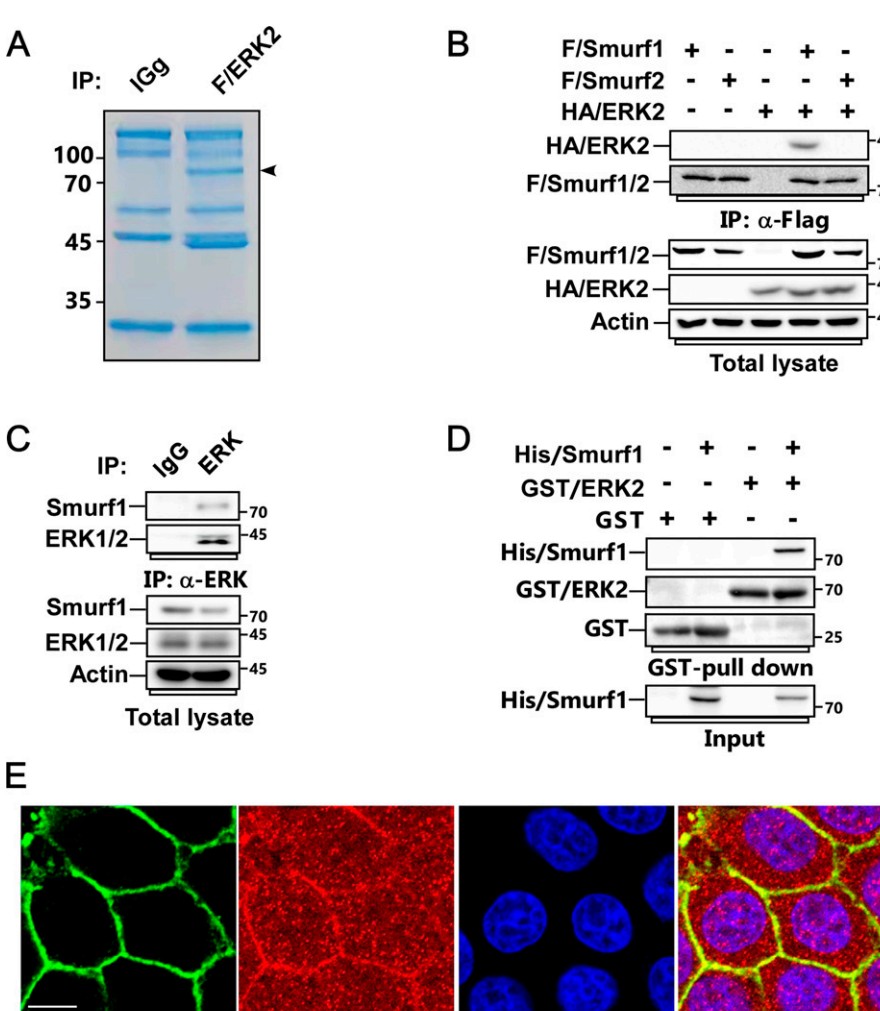

**Figure 1. Smurf1 is a new binding partner of ERK.**
**(A)** MCF-7 cells transfected with Flag-tagged ERK2 were subjected to anti-Flag immunoprecipitation (IP), followed by SDS–PAGE and Coomassie brilliant blue staining. The indicated band was analyzed by mass spectrometry. **(B)** MCF-7 cells transfected with FLAG-tagged Smurf1-C699A (FLAG/Smurf1-C699A) or Smurf2-C716A (FLAG/Smurf2-C716A) and HA-tagged ERK2 were subjected to anti-Flag IP and immunoblot (IB) to detect the interaction between FLAG/Smurfs and HA/ERK2. **(C)** Cell lysates from MCF-7 cells were subjected to anti-ERK IP followed by IB to detect the associated Smurf1. **(D)** GST-tagged ERK2 and His-tagged Smurf1 purified from bacteria were subjected to GST-Pull down assay followed by IB to detect their interaction. **(E)** MCF-7 cells expressing FLAG/Smurf1-C699A were subjected to immunofluorescence (IF) assay to detect the colocolization of FLAG/Smurf1-C699A and endogenous ERK. Scale bar, 10 μm.

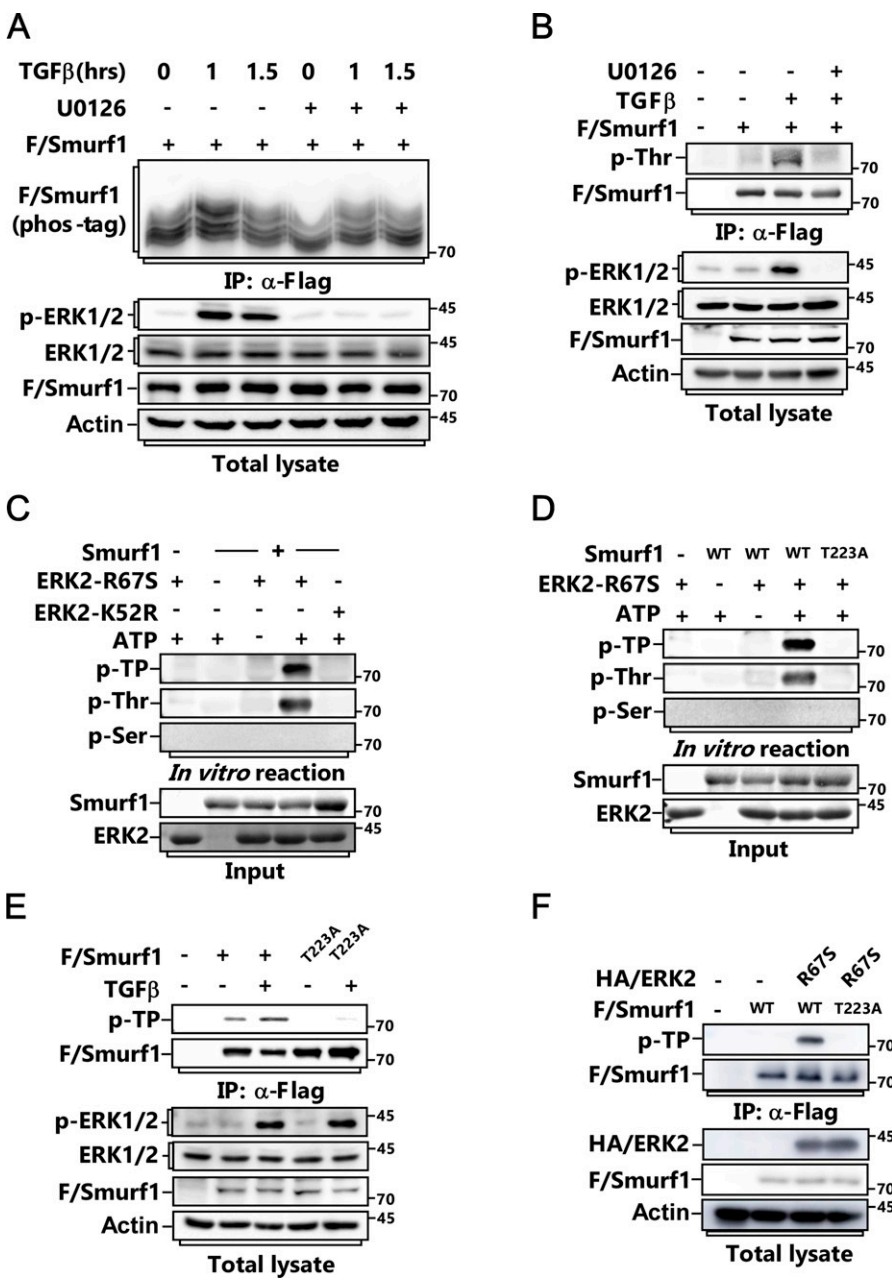

**Figure 2. ERK phosphorylates Smurf1 on Threonine 223.**
**(A)** MCF-7 cells transfected with FLAG/Smurf1 were pretreated with or without 5 μM U0126 for 2 h before being treated with or without TGFβ for the indicated time and subjected to anti-Flag IP followed by phospho-tag assay and IB to detect the phosphorylation of FLAG/Smurf1. **(B)** MCF-7 cells transfected with FLAG/Smurf1 were pretreated with or without 5 μM U0126 for 2 h before being treated with or without TGFβ for another 1 h and subjected to anti-Flag IP followed by IB using phospho-threonine antibody to detect the phosphorylation of FLAG/Smurf1. **(C)** In vitro kinase assay was carried out by incubating bacterially expressed and purified Smurf1 with constitutively active form of ERK2 (ERK2-R67S) or catalytically inactive form of ERK2 (ERK2-K52R). Phosphorylated Smurf1 was detected by IB using phospho-threonine-proline, phospho-threonine, or phospho-serine anti-bodies. **(D)** Bacterially expressed and purified Smurf1-WT or Smurf1-T223A mutant was subjected to in vitro kinase assay with ERK2-R67S. Phosphorylated Smurf1 was detected by IB using phospho-threonine-proline, phospho-threonine, or phospho-serine anti-bodies. **(E)** MCF-7 cells transfected with FLAG/Smurf1-WT or FLAG/Smurf1-T223A mutant were treated with or without TGFβ for 1 h and subjected to anti-Flag IP followed by IB using phospho-threonine-proline antibody to detect the phosphorylation of FLAG/Smurf1. **(F)** MCF-7 cells transfected with FLAG/Smurf1-WT or FLAG/Smurf1-T223A mutant and HA-tagged ERK2-R67S (HA/ERK2-R67S) were subjected to anti-Flag IP followed by IB using phospho-threonine-proline antibody to detect the phosphorylation of FLAG/Smurf1.

inactive form of ERK2 (ERK2-K52R) and discovered that ERK2-R67S could phosphorylate Smurf1 on Threonine but not on serine residue(s) using phospho-Thr- and phospho-ser-specific antibodies (Fig 2C). Meanwhile, ERK2-R67S but not ERK2-K52R phosphorylated Smurf1 in vivo (Fig S2D). To identify the phosphorylation site(s), we carried out matrix-assisted laser desorption/ionization time-of-flight mass spectrometry (MALDITOF-MS) analysis after kinase reaction and found that T223 of Human Smurf1 was phosphorylated by ERK, which is very conserved in different kinds of species (Fig S2E). Accordingly, mutation of this residue to alanine totally abolished TGFβ-induced and ERK-mediated phosphorylation of Smurf1 both in vivo and in vitro (Figs 2D–F and S2F and G).

## ERK-mediated Smurf1 phosphorylation is necessary for TGFβ-induced RhoA degradation

Because both ERK and Smurf1 are involved in EMT (Fan et al, 2019; Olea-Flores et al, 2019; Fu et al, 2020; Wu et al, 2020), meanwhile, Smurf1-mediated degradation of RhoA and the consequent cortical actin filaments disassembly are critical for TGFβ-induced EMT (Ozdamar et al, 2005; Sanchez & Barnett, 2012). We hypothesized that ERK-mediated Smurf1 phosphorylation might have an essential role in this process. As predicted, MEK inhibitor U0126 but not PI3K inhibitor LY294002 or AKT inhibitor MK2206 blocked TGFβ-induced RhoA degradation (Figs 3A and B and S3A–C). Furthermore, knockdown of ERK could also attenuate RhoA degradation (Fig S3D).

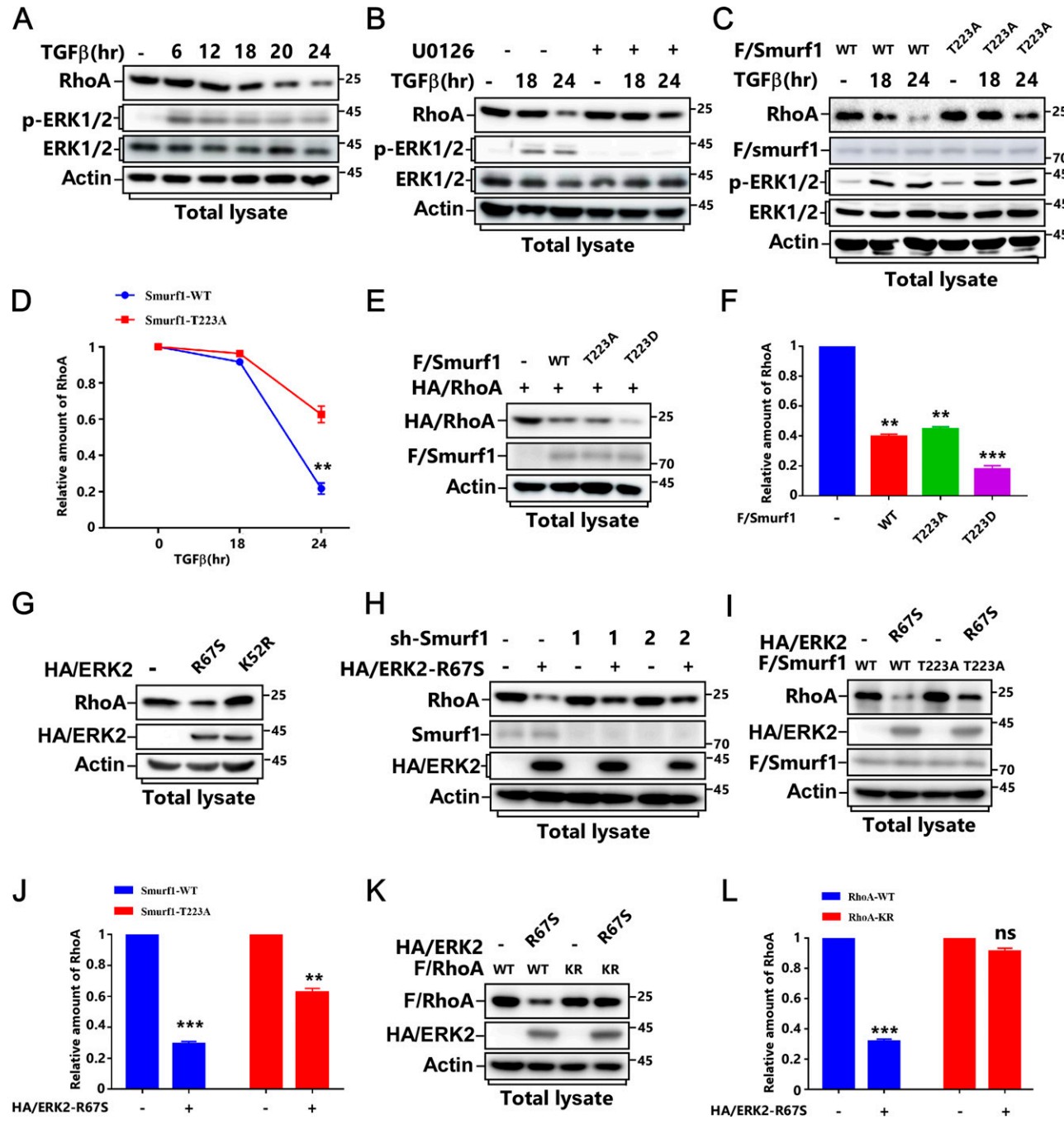

**Figure 3. T223 phosphorylation of Smurf1 is essential for RhoA degradation after TGFβ treatment.**
**(A)** MCF-7 cells were treated with or without TGFβ for the indicated time and subjected to IB to detect RhoA protein levels. **(B)** MCF-7 cells were pretreated with or without 5 µM U0126 for 2 h before being treated with or without TGFβ for the indicated time and subjected to IB to detect RhoA protein levels. **(C, D)** MCF-7 cells transfected with FLAG/Smurf1-WT or FLAG/Smurf1-T223A mutant were treated with or without TGFβ for the indicated time and subjected to immunoblot IB to detect RhoA protein levels (C). The quantified data were plotted as mean ± SD of three independent experiments. **(D)** **$P < 0.01$ (one-way ANOVA with LSD post hoc test) (D). **(E, F)** MCF-7 cells transfected with FLAG/Smurf1-WT, FLAG/Smurf1-T223A, or FLAG/Smurf1-T223D mutant and HA/RhoA were subjected to IB to detect HA/RhoA protein levels (E). The quantified data were plotted as mean ± SD of three independent experiments. **(F)** **$P < 0.01$; ***$P < 0.001$ (one-way ANOVA with LSD post hoc test) (F). **(G)** MCF-7 cells transfected with HA/ERK2-R67S or HA/ERK2-K52R were subjected to IB to detect RhoA protein levels. **(H)** MCF-7 cells transfected with lentivirus encoding HA/ERK2-R67S and con-shRNA (sh-Con) or sh-RNA against Smurf1 (sh-Smurf1-1 or 2) were subjected to IB to detect RhoA protein levels. **(I, J)** MCF-7 cells transfected with FLAG/Smurf1-WT or FLAG/Smurf1-T223A mutant and HA/ERK2-R67S as indicated were subjected to IB to detect RhoA protein levels (I). The quantified data were plotted as mean ± SD of three independent experiments. **(J)** **$P < 0.01$; ***$P < 0.001$ (one-way ANOVA with LSD post hoc test) (J). **(K, L)** MCF-7 cells transfected with FLAG/RhoA-WT or FLAG/RhoA-K6, 7R mutant, and HA/ERK2-R67S as indicated were subjected to IB to detect RhoA protein levels (K). The quantified data were plotted as mean ± SD of three independent experiments. **(L)** ***$P < 0.001$; NS, not significant (one-way ANOVA with LSD post hoc test) (L).

We next examined whether Smurf1 phosphorylation was re-quired for TGFβ-induced RhoA degradation. TGFβ treatment caused RhoA degradation in Smurf1-WT-expressing cells. However, Smurf1-T223A mutation or knockdown of Smurf1 notably blocked this biological process (Figs 3C and D and S3E and F). Accordingly, Smurf1-T223D could further reduce RhoA protein levels than that of Smurf1-WT and Smurf1-T223A (Fig 3E and F).

To verify ERK and Smurf1 are in the same pathway to regulate RhoA turnover, we checked RhoA protein levels in ERK2-overexpressing cells and discovered that introduction of ERK2-R67S but not ERK2-K52R led to RhoA down-regulation, whereas knockdown of Smurf1 abolished RhoA reduction (Fig 3G and H). Accordingly, both Smurf1-T223A and RhoA-KR, which blocks smurf1-mediated RhoA ubiquitination, notably attenuated ERK2-R67S-induced RhoA turnover (Fig 3I–L). Meanwhile, Smurf1-T223A markedly improved RhoA stability (Fig S3G and H), and the proteasome inhibitor MG-132 could effectively block ERK2-R67S-induced RhoA down-regulation (Fig S3I). Thus, ERK phosphory-lates Smurf1 to promote RhoA ubiquitination and degradation.

### ERK1/2 phosphorylates Smurf1 to promote its binding to RhoA and subsequent ubiquitination

Next, we wanted to know the mechanism underlying Smurf1 phosphorylation influences RhoA degradation. We observed that the interaction between Smurf1 and RhoA was markedly enhanced after TGFβ treatment, whereas this could be totally blocked by ERK inhibitor U0126 or Smurf1 phosphorylation mutant Smurf1-T223A (Fig 4A and B). In good agreement with this, Smurf1-T223D obviously enhanced, whereas Smurf1-T223A reduced, the binding of Smurf1 to RhoA (Fig 4C). Moreover, we conducted GST pull-down assay after kinase reaction, as shown in Fig 4D; the interaction between RhoA and Smurf1-WT but not Smurf1-T223A was notably enhanced by ERK-mediated phosphorylation in vitro. Furthermore, we performed protein–protein Molecular Docking Simulation, and found that phosphorylation of Smurf1 on Threonine 223 could lead to salt bridge formation between pThr223 of Smurf1 and His105 of RhoA (Fig S4A), resulting in the enhanced interaction.

Accordingly, we carried out ubiquitination assay, and discovered that TGFβ treatment led to polyubiquitination of RhoA, and this could also be attenuated by U0126 (Fig 4E). Meanwhile, both constitutively active form of ERK2 (ERK2-R67S) and Smurf1-T223D could promote, but Smurf1-T223A could impair, RhoA poly-ubiquitination (Figs 4F and S4B). Thus, phosphorylation of Smurf1 is necessary for its sufficient binding to RhoA and the consequent RhoA ubiquitination and degradation.

### Phosphorylation of Smurf1 is required for EMT and breast cancer metastasis

To determine the biological function of ERK-mediated Smurf1 phosphorylation both in vitro and in vivo, we pretreated MCF-7 cells with MEK inhibitor U0126 and then treated the cells with TGFβ, and noted that U0126 markedly blocked TGFβ-induced EMT, as indicated by the epithelial marker E-cadherin and ZO-1 (Fig 5A). In good agreement, overexpression of ERK-R67S but not ERK-K52R was able to induce EMT (Fig S5A). Next, we reintroduced Smurf1-WT or

Smurf1-T223A into Smurf1 knockdown cell line and then treated the cells with TGFβ. Smurf1-T223A mutant attenuated TGFβ-induced EMT (Fig 5B). Accordingly, Smurf1-T223A mutant significantly re-duced cell migration and invasion (Fig 5C and D). Then we gen-erated Smurf1-WT or Smurf1-T223A expressing breast cancer cells by reintroducing Smurf1-WT or Smurf1-T223A into Smurf1 knock-down cells, and injected the cells into the mammary fat pad of female BALB/c mice to examine the primary tumor growth and lung metastasis. The phosphorylation of Smurf1 had no significant effect on primary tumor growth (Fig S5B). Accordingly, phosphorylation-resistant mutant of Smurf1 did not influence cell proliferation or apoptosis of the primary tumor (Fig S5C). However, it dramatically attenuated the lung metastasis of the breast cancer cells (Figs 5E and F and S5D), confirming that Smurf1 phosphorylation is required for EMT and tumor metastasis.

## Discussion

TGFβ-induced EMT, which undergoes through both smad-dependent and smad-independent pathways, provides funda-mental roles in physiological and pathological processes. Previous study revealed that, in the early stage of EMT, TGFβ treatment leads to TβRII activation and the partitioning-defective protein 6 (Par6) phosphorylation. Phosphorylated Par6 recruits Smurf1 to the TJ region, where Smurf1 catalyzes localized RhoA for degradation, resulting in dissolution of cortical actin (Ozdamar et al, 2005; Sanchez & Barnett, 2012). Meanwhile, TGFβ treatment promotes ERK1/2 activation followed by p120 phosphorylation, which en-hances the interaction between p120 and smurf1. Smurf1, thereby, catalyzes p120 mono-ubiquitination, leading to p120 dissembles from cadherin complex and finally AJ disruption (Wu et al, 2020).

Multiple extracellular cues could catalyze EMT, among all of these factors TGFβ signaling bifurcating at many points including MAPK pathway, PI3K-AKT and RhoA pathways plays pivotal roles (Derynck & Zhang, 2003; Xie et al, 2004; Moustakas & Heldin, 2005; Xu et al, 2009). Consistent with previous studies, we identified ERK, including both ERK1 and ERK2, as key kinases regulating EMT in a non-transcriptional way. Kinase assay implicates that ERK phos-phorylates Smurf1 at T223, leading to its enhanced interaction with RhoA and then RhoA ubiquitination. Degradation of localized RhoA is critical for actin cytoskeleton remodeling and cell–cell junction dissociation during EMT.

Smurf1 could regulate cell polarity and cellular protrusion for-mation by targeting RhoA for degradation in filopodia and lamellipodia (Wang et al, 2003, 2006). Meanwhile, Smurf1-catalyzed localized RhoA turnover is essential for TGFβ-induced EMT (Ozdamar et al, 2005). However, it is still unclear how Smurf1 is regulated in this biological process. Here we uncovered that Smurf1 could be phosphorylated by ERK, and the phosphorylation of Smurf1 on T223 identified in this study presents a linkage between Smurf1 and RhoA, revealing a mechanism of how Smurf1 is regu-lated to promote RhoA degradation. Thus, Smurf1-targeted RhoA degradation could be regulated in two steps. On one hand, TβRII-mediated Par6 phosphorylation attracts Smurf1 to TJ region. On the other hand, ERK phosphorylates Smurf1 to enhance its binding to

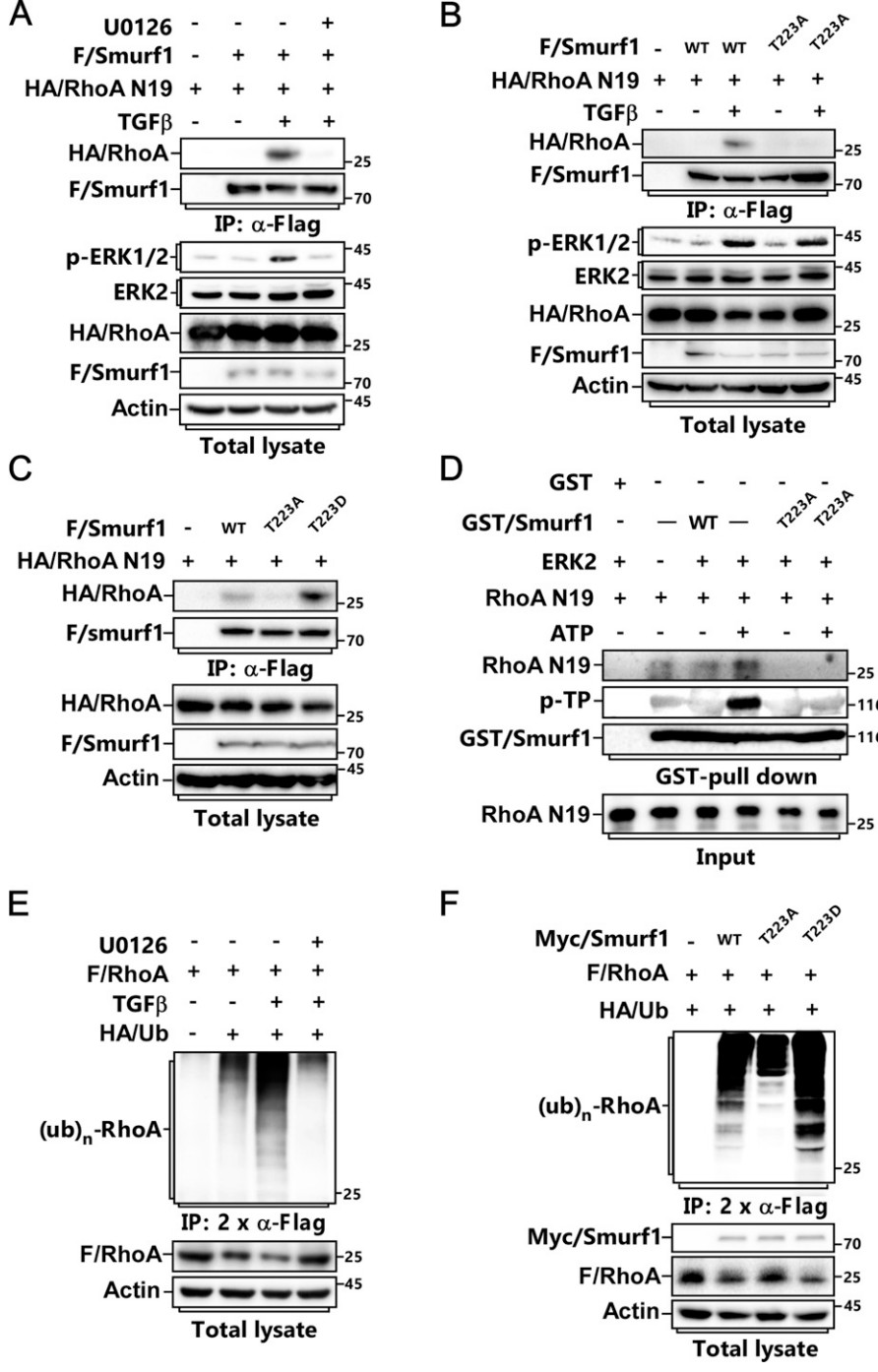

**Figure 4. T223 phosphorylation of Smurf1 is essential for its interaction with RhoA and subsequent ubiquitination.**

**(A)** MCF-7 cells transfected with FLAG/Smurf1 and HA/RhoA N19 were pretreated with or without 5 μM U0126 for 2 h before being treated with or without TGFβ for 2 h and subjected to anti-Flag IP and IB to detect the associated HA/RhoA N19. **(B)** MCF-7 cells transfected with FLAG/Smurf1-WT or FLAG/Smurf1-T223A mutant and HA/RhoA N19 were treated with or without TGFβ for 2 h and subjected to anti-Flag IP and IB to detect the associated RhoA N19. **(C)** MCF-7 cells transfected with FLAG/Smurf1-WT, FLAG/Smurf1-T223A or FLAG/Smurf1-T223D mutant and HA/RhoA N19 were subjected to anti-Flag IP and IB to detect the associated HA/RhoA N19. **(D)** GST/Smurf1-WT or GST/Smurf1-T223A purified from bacteria was incubated with ERK2-R67S for kinase assay and then subjected to GST-Pull down assay with RhoA N19 followed by IB to detect the associated RhoA N19. **(E)** MCF-7 cells transfected with FLAG/RhoA and HA/Ub were pretreated with or without 5 μM U0126 for 2 h before being treated with or without TGFβ for 24 h and subjected to ubiquitination assay to detect FLAG/RhoA ubiquitination. **(F)** MCF-7 cells transfected with FLAG/RhoA, HA/Ub, and Myc/Smurf1-WT, Myc/Smurf1-T223A, or Myc/Smurf1-T223D were subjected to ubiquitination assay to detect FLAG/RhoA ubiquitination.

RhoA and promote RhoA degradation. However, whether these two affairs are concomitant events or sequential events still needs our further efforts. Besides, we observed that the levels of T223 phosphorylation of Smurf1 increase and then decrease after TGFβ treatment, suggesting a dynamic regulation of Smurf1 phosphorylation during EMT through a yet unknown mechanism.

Numerous researches suggest a pivotal role of Smurf1 in cancer progression, including breast cancer, colon cancer, and pancreatic cancer (Suzuki et al, 2008; Birnbaum et al, 2011; Nie et al, 2016; Wu

et al, 2020). Meanwhile, the activity of Smurf1 in mediating RhoA degradation is critical for cancer cell invasive activity (Kwei et al, 2011; Kwon et al, 2013). Recently, Fan et al demonstrated that Smurf1 could promote ovarian cancer cell EMT in a RhoA-independent way (Fan et al, 2019). They revealed that the absence of Smurf1 represses cell proliferation, invasive capability, and EMT process in ovarian cancer through DAB2IP/AKT/Skp2 signaling loops. Thus, Smurf1 plays important roles in cancer progression in different ways depending on different context.

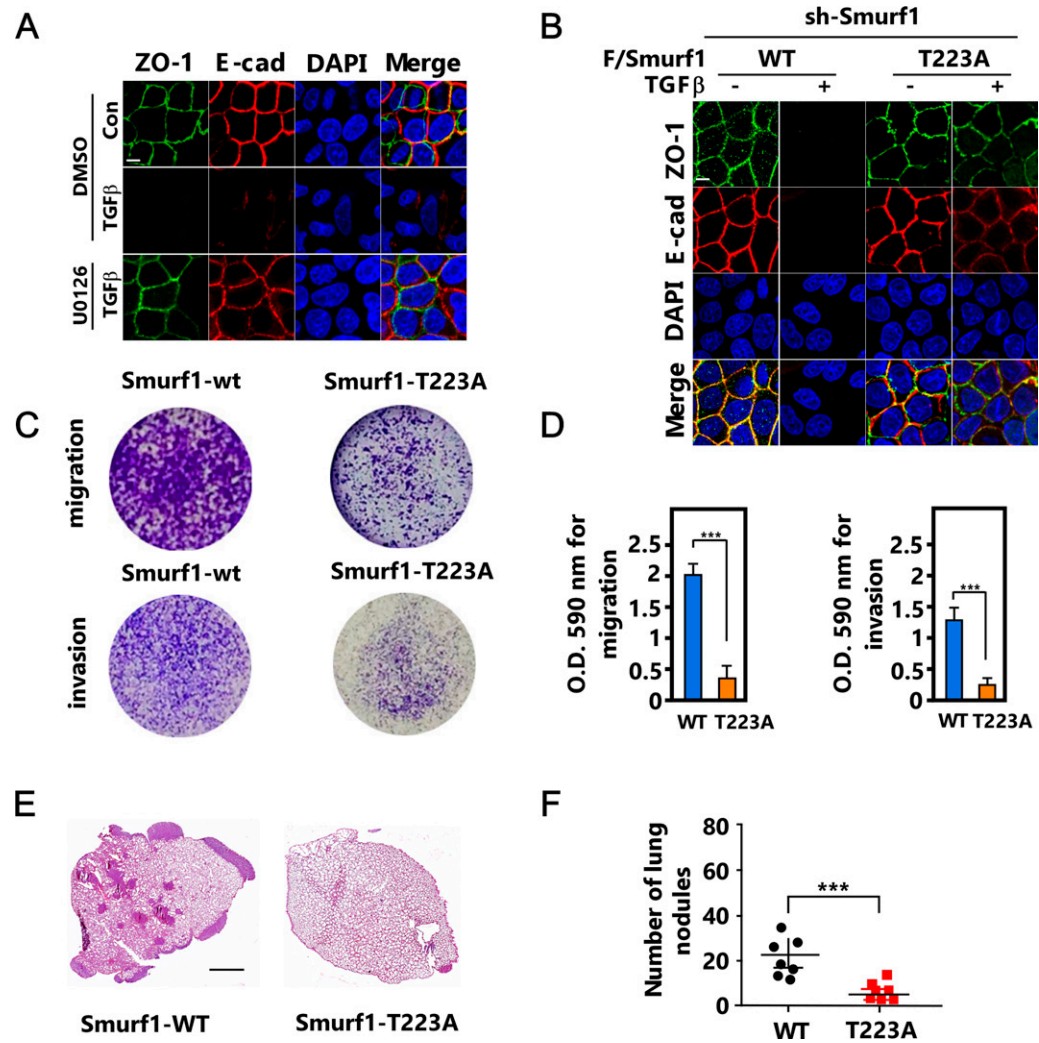

**Figure 5. T223 phosphorylation of Smurf1 is essential for epithelial-mesenchymal transition and breast cancer metastasis.**
**(A)** MCF-7 cells were pretreated with or without 5 $\mu$M U0126 for 2 h before being treated with or without TGF$\beta$ for 24 h and subjected to immunofluorescence (IF) assay to detect cell-cell junctions. Scale bar, 10 $\mu$m. **(B)** Smurf1-knockdown MCF-7 cells transfected with FLAG/Smurf1-WT or FLAG/Smurf1-T223A were treated with or without TGF$\beta$ for 24 h and subjected to immunofluorescence (IF) assay to detect cell-cell junctions. Scale bar, 10 $\mu$m. **(C, D)** Smurf1-knockdown MDA-MB-231 cells transfected with FLAG/Smurf1-WT, or FLAG/Smurf1-T223A were subjected to migration or invasion assay. **(C)** Representative images of migrating and invading cells stained with crystal violet (C). The histograms are the quantitation of the migrating and invading cells, the quantified data were plotted as mean ± SD of three independent experiments. **(D)** ***P < 0.001 (one-way ANOVA with LSD post hoc test) (D). **(E, F)** Smurf1-knockdown 4T1 cells transfected with FLAG/Smurf1-WT, or FLAG/Smurf1-T223A (0.5 × 10$^6$) were orthotopically injected into the mammary fat pad of female BALB/c mice (n = 7 mice per group). For metastasis assays, tumors were surgically resected when they reached a volume greater than 300 mm$^3$, 25 d after injection the mice were euthanized. Hematoxylin and eosin (H&E)–stained lung sections; scale bar, 1 mm. **(E)** The numbers of lung metastasis nodules were counted and presented as mean ± SD of seven mice per group. **(F)** ***P < 0.001 (one-way ANOVA with LSD post hoc test) (F).

Altogether, our data uncovered a detailed unknown mechanism during EMT and paved the way for deeply understanding the mechanisms underlying tumor invasion and metastasis, providing new clues for therapeutic intervention.

# Materials and Methods

### DNA constructs

The cDNAs of human Smurf1, Smurf2, ERK1, and ERK2 were generous gifts from Dr. H-RW. Mutations of Smurfs and ERK1/2 were generated by PCR-based site-directed mutagenesis. Cloning for protein expression in mammalian cells was carried out using a pCMV6 vector for transfection, pBOBI for lentivirus infection. pGEX-4T-1 and pRroEX were used for bacterial expression of proteins. Human Smurf1 (wild-type and C699A) and Smurf2 (wild-type and C716A) have been previously reported (Nieto et al, 2016). The lentiviral-based vectors pLL3.7 were used for shRNA expression. The sequences used in MCF-7 cells for expression of for ERK1 shRNA is 5′-GCATTCTGGCTGAGATGCTCT-3′; for ERK2 shRNA is 5′-GCGCTTCAGACATGAGAAC-3′; and for Smurf1 shRNA-1 and shRNA-2 are 5′-TATTCTACGGACAACATTT-3′ and 5′-GATAGGCACTGGAGGCTCTGT-3′, respectively. The scramble sequence 5′-TTCTCCGAACGTGGCACGA-3′ was used for a control shRNA.

## Antibodies and chemical reagents

Mouse anti–E-Cadherin (1:1,000, Cat. no. 14472s), Rabbit anti–ERK1/2 (1:2,000, Cat. no. 9102s), anti–phospho-ERK1/2 (1:2,000, Cat. no. 4370T), anti–Akt(pan) (C67E7) (1:2,000, Cat. no. 4691s), anti–phospho-Akt(ser473) (193H12) (1:2,000, Cat. no. 4058), and anti–phospho-threonine (1:1,000, Cat. no. 9381) were purchased from Cell Signaling Technology; mouse anti-Myc (1:2,000, Cat. no. sc-40), anti-GST(B-14) (1:2,000, Cat. no. sc-138), and anti-Actin(C4) (1:2,000, Cat. no. sc-47778) were purchased from Santa Cruz Biotechnology; mouse anti-Smurf1 (1:2,000, Cat. no. ab57573), Rabbit anti-Ki67 (1:100, Cat. no. ab15580), Rabbit anti–Cleaved caspase-3 (1:100, Cat. no. ab32351), and Goat anti-rabbit IgG H&L(HRP) (1:5,000, Cat. no. ab27236), anti-mouse IgG H&L(HRP) (1:5,000, Cat. no. ab27241), and anti-rat igG H&L(HRP) (1:5,000, Cat. no. ab97057) were purchased from Abcam; Alexa Fluor (R) 555 donkey anti-mouse (1:500, Cat. no. A31570) and Alexa Fluor (R) 488 donkey anti-rat (1:500, Cat. no. A21208) were purchased from Thermo Fisher Scientific; mouse anti-Flag (M2) (1:2,000, Cat. no. F1804) was purchased from Sigma-Aldrich; rat anti–ZO-1(1:2,000, Cat. no. MABT11) and mouse anti-phosphoserine (1:1,000, Cat. no. 05-1000) were purchased from Merck Millipore; rat anti-HA (1:2,000, Cat. no. 11867423001) was purchased from Roche; inhibitors for MEK U0126-EtOH (Cat. no. HY-12031), inhibitors for AKT MK-2206 dihydrochloride (Cat. no. 1032350-13-2), and inhibitors for PI3K LY294002 (Cat. no. HY-10108) were purchased from Med-ChemExpress (MCE); diamidino-2-phenylindole dihydrochloride (DAPI) (Cat. no. D1306) was purchased from Thermo Fisher Scientific; and N-ethylmaleimide (NEM) (Cat. no. A600450) was purchased from Sangon Biotech.

## Cell culture and TGFβ treatment

Human breast cancer MCF-7, MDA-MB-231, and mouse breast cancer 4T1 were purchased from ATCC. MCF-7 and MDA-MB-231 cells were cultured in high-glucose DMEM, 4T1 was cultured in RPMI-1640, all supplemented with 10% (vol/vol) FBS (Thermo Fisher Scientific) and 100 units/ml streptomycin and penicillin (Millipore), at 37°C in a humidified 5% $CO_2$ incubator. The cell lines were routinely tested and found negative for mycoplasma. For TGFβ treatment, the cells were washed with PBS, cultured in DMEM with 0.05% FBS, and then treated with TGFβ for determined time.

## Transfection, generation of the lentivirus, and infection

Plasmids transient transfection was performed using Lipofect-amine 2000 (Invitrogen) following the manufacturer's protocol. For lentivirus production, HEK293T cells were transfected with indicated plasmids in 100-mm dishes; 12 h after transfection fresh DMEM medium was changed. 2 d after medium changing, viral supernatants were collected and centrifuged at 70,000*g* for 3 h, resuspended, filtered through 0.45-μm filters, and then stored in –80°C. The cells were infected with lentiviruses supplemented with polybrene for 12 h and selected with puromycin for at least 2 d.

## Immunoprecipitation and GST pull-down assay

Immunoprecipitation (IP) and GST pull-down assays were performed as previously described. Briefly, cells were lysed on ice with lysis buffer TNTE 0.5% (50 mM Tris–HCl, pH 7.5, 150 mM NaCl, 1 mM EDTA, and 0.5% Triton X-100, containing 10 mg ml$^{-1}$ pepstatin A, 10 mg ml$^{-1}$ leupeptin, and 1 mM PMSF). The cell lysates were applied to IP assays with appropriate antibodies. For GST pull-down assay, bacterially expressed GST/ERK2 and GST/smurf1 were purified using glutathione sepharose beads in TNTE 0.5% buffer, bacterially expressed His/smurf1 was purified using nickel beads in TNT 0.5% (50 mM Tris–HCl, pH 7.5, 150 mM NaCl, 0.5% Triton X-100, containing 10 mg ml$^{-1}$ pepstatin A, 10 mg ml$^{-1}$ leupeptin, and 1 mM PMSF).

## Ubiquitination assay

Ubiquitination assay also performed as described previously. For in vivo ubiquitylation assay, cell lysates were subjected to anti-Flag IP for 3 h, eluted by boiling 5 min in 1% SDS, diluted 10 times in lysis buffer TNTE 0.5%, and then re-immunoprecipitated with anti-Flag (2×IP) for 12 h. The ubiquitin-conjugated proteins were detected by WB.

## In vitro kinase assay

For in vitro kinase assay, ERK1/2 and Smurf1 were bacterially expressed. The indicated proteins were incubated in kinase buffer (20 mM Tris HCl, pH 7.5, 10 mM $MgCl_2$, 1 mM dithiothreitol, and 25 μM ATP) at 37°C for 1 h followed by WB.

## Immunofluorescence assay

After indicated treatments, cells grown on glass coverslips were washed three times in PBS, fixed with 4% PFA and permeabilized with 0.25% Triton X-100, then stained using appropriate primary (Flag (1:100; Sigma-Aldrich), E-cad (1:100; Sigma-Aldrich), ZO-1(1:100; BD)) and proper fluorescently conjugated secondary antibodies (1:500; Invitrogen). Images were obtained using a ZEISS LSM 780 confocal microscope with ZEN 2010 software (Carl Zeiss GmbH).

## Histological assays

Tissues were fixed in 10% formalin overnight, dehydrated, and embedded in paraffin and paraffin-embedded tissue samples were sectioned, deparaffinized, rehydrated. For lung metastasis assay, the slides were stained with hematoxylin and eosin (H&E) followed by washing with $H_2O$. For immunohistochemistry assay of the primary tumors, the slides were boiled for 30 min in sodium citrate/citric acid mixture (pH 6.0) for antigen retrieval and then pretreated with peroxidase blocking buffer (Maxim) for 20 min at room temperature. The slides were incubated with appropriate primary antibodies overnight at 4°C after being blocked with 5% normal goat serum for 1 h. The UltraSensitive SP kit (Maxim) was then used to detect the specific primary antibodies.

## Cell migration and invasion assays

The cell migration and invasion assays were performed in a 24-well Transwell plate with 8-$\mu$m polyethylene terephthalate membrane filters. MDA-MB-231 cells were plated in the upper chamber at $0.5 \times 10^5$ cells per well in serum-free DMEM, whereas the bottom chamber contained DMEM with 10% FBS. Cells were allowed to migrate for 5 h, and the migrated cells were counted after fixation and staining. The invasion assay was similar to the migration assay, except that the membrane filter was precoated with diluted Matrigel before the assay and the incubation time was 15 h.

## Animal studies

Female BALB/c mice (6 wk old) were purchased from and housed in Laboratory Animal Center of Xiamen University in a facility with 12-h light/12-h dark cycles under pathogen-free conditions. Mouse experiments were performed in accordance with protocols approved by the Institutional Animal Care and Use Committee of Xiamen University. Smurf1 knockdown cells with reintroduction of Smurf1-WT or Smurf1-T223A ($0.5 \times 10^6$) were harvest in PBS (40 $\mu$l) and injected into the mammary fat pad of the mice. For primary tumorigenicity assay, the mice were euthanized 25 d after injection and the primary tumors were weighted. For the lung metastasis experiments, tumors were surgically resected when they reached a volume greater than 300 mm$^3$ and the mice were killed 25 d after injection and the lung metastasis colonies was counted.

## Statistical analysis

One-way ANOVA with LSD post hoc test was used to compare values among different experimental groups using the GraphPad prism program version 6.01. $P < 0.05$ was considered a statistically significant change. $*P < 0.05$; $**P < 0.01$; $***P < 0.001$; NS, not significant. All the values were presented as mean ± SD of at least triplicate experiments.

# Data Availability

All data supporting the findings of this study are available within the article and its Supplementary Information files or from the corresponding author upon request.

# Supplementary Information

# Acknowledgements

This work was supported by Natural Science Foundation of China grants (No. 81972373), the funding of Science and Technology Planned project of Medical and Health of Xiamen City (3502Z20194038), Scientific Research Foundation for Advanced Talents, Xiang'an Hospital of Xiamen University (NO. PM201809170001), and the Natural Science Foundation of Fujian Province of China (No. 2019J01016).

## Author Contributions

J Zheng: conceptualization, data curation, software, formal analysis, funding acquisition, investigation, and methodology.
Z Shi: data curation, software, and formal analysis.
P Yang: resources and software.
Y Zhao: software, resources, and data curation.
W Tang: resources.
S Ye: resources.
Z Xuan: software and formal analysis.
C Chen: software and methodology.
C Shao: supervision, funding acquisition, project administration, and writing—original draft, review, and editing.
Q Wu: supervision, project administration, and writing—original draft, review, and editing.
H Sun: supervision, funding acquisition, and writing—original draft.

## Conflict of Interest Statement

The authors declare that they have no conflict of interest.

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
