## [Reviewer comments · Life Science Alliance]

Life Science Alliance

ERK-Smurf1-RhoA signaling is critical for TGF-beta-driven EMT and tumor metastasis

Jianzhong Zheng, Yue Zhao, Pengbo Yang, Zhiyuan Shi, Wenbin Tang, Shaopei Ye, Zuodong Xuan, Chen Chen, Chen Shao, Qingang Wu, and Huimin Sun

DOI: <https://doi.org/10.26508/lsa.202101330>

Corresponding author(s): Dr. Qingang Wu (First Affiliated Hospital Zhejiang University)

Review Timeline:

Submission Date:	2021-12-08
Editorial Decision:	2022-01-03
Revision Received:	2022-04-18
Editorial Decision:	2022-05-09
Revision Received:	2022-05-16
Accepted:	2022-05-17

Transaction Report:

January 3, 2022

Re: Life Science Alliance manuscript #LSA-2021-01330-T

Dr. Qingang Wu

Zhejiang Provincial Key Laboratory of Pancreatic Disease, The First Affiliated Hospital, and Institute of Translational Medicine, Zhejiang University School of Medicine
China

Dear Dr. Wu,

Thank you for submitting your manuscript entitled "ERK-Smurf1-RhoA signaling is critical for TGF-beta-driven EMT and tumor metastasis" to Life Science Alliance. The manuscript was assessed by expert reviewers, whose comments are appended to this letter. We, thus, encourage you to submit a revised version of the manuscript back to LSA that responds to all of the reviewers' points.

Thank you for this interesting contribution to Life Science Alliance. We are looking forward to receiving your revised manuscript.

Sincerely,

B. MANUSCRIPT ORGANIZATION AND FORMATTING:

Reviewer #1 (Comments to the Authors (Required)):

The critical function of MEK-ERK1/2 signaling in tumor initiation, progression and metastasis formation has been documented. Here, Zheng and co-workers have investigated the functional contribution of ERK1/2 signaling to EMT in MCF7 breast cancer cells and on tumor metastasis in the 4T1 syngeneic transplantation mouse model of breast cancer. Notably, they have set out to identify novel non-canonical pathways of ERK1/2 signaling. They report the identification of Smurf1 as a binding partner of ERK1/2 and that ERK1/2-mediated phosphorylation of Smurf1 at Thr223 promotes Smurf1-mediated poly-ubiquitination and degradation of RhoA, thus promoting EMT. Blockade of ERK1/2-mediated phosphorylation of Smurf1 and of RhoA degradation represses TGF β -induced EMT and experimental breast cancer metastasis.

The study reports novel insights into an ERK1/2 pathway, thus far neglected in EMT and metastasis research. The experimental approaches are thoughtfully designed, yet they are limited to biochemical and cellular analyses on almost exclusive MCF7 cells only. The results are carefully interpreted, yet may lack generality. Overall, this is an interesting and medically relevant report which seems still preliminary and in parts limited in its proof-of-concept experimentation. Also, a previous manuscript has reported a critical role of Smurf1 in EMT of ovarian cancer cells (Ref. 31). The authors should specifically discuss their results in the context of this recent publication.

Specific comments:

1. The selection of general reviews referred to in the Introduction is curious and rather not specific to the actual topic: For example, reference 23 deals with ERK signaling and depression....
2. English Grammar and Style are not of high quality and the manuscript certainly needs adequate proofreading.
3. Figure 1: It seems important to show a co-immunoprecipitation of both endogenous proteins, ERK1/2 and Smurf1. In the data shown, one protein is always overexpressed and tagged.
4. In the experiments shown in Figure 1, it is not clear whether the cells have been treated with TGF β . If not, this would mean that ERK1/2 are already interacting with Smurf1 without activation of ERK1/2? Later in the manuscript in Figure 4A and B, it is shown that they interact only upon TGF β stimulation. What is then the mechanisms that allows the interaction only after TGF β stimulation, changes in localization, in ERK1/2 activity or in the recruitment of co-factors or else?
5. Figure S2E is shown before Figure S2C (order of appearance).
6. The authors argue that the loss of RhoA disrupts the Par6 polarity complexes and thus dissolves cortical actin. On the other hand, RhoA is a well-known player in the formation of stress fibers, which in turn are required for EMT and cell migration. How can the cells undergo EMT, when RhoA is lost?
7. The number of repeats of each experiment should be mentioned in the figure legends. Some immunoblots are not of adequate quality to allow a quantitative interpretation (for example Figure 3D, F, H), better blots should be shown and actually quantified.
8. The immunoblots shown in Figure 4E and F and S4A are not clear: Ubiquitinated proteins usually run as ladders above the non-ubiquitinated forms, hence the ladder of proteins immunoblotted for ubiquitin should go from bottom to top, not the other way around as shown here?
9. The proteasomal degradation of RhoA, as claimed by the authors, should be demonstrated for example by using a proteasomal inhibitor or by time course and pulse chase experiments to assess RhoA protein stability.
10. Figure 5E and F: The bioluminescence images and their quantification are not clear: the primary tumors seem to have been covered in the bioluminescence imaging process, since their high activities would have impaired the analysis of the metastatic lesion. Still, without removal of the primary tumor, how can this be adequately quantified? There also seem to be metastatic lesions in the brain and less in the lung, the latter known to be the major target organ of 4T1 cells. It may be advised to remove

the primary tumors when reaching a certain size and then assess metastasis formation. Along these lines, it may be important to learn whether the Smurf1 phosphorylation-deficient tumors show any difference in histopathological parameters, such as invasion, proliferation, apoptosis etc.

Furthermore, it is not clear how the Smurf1-WT and T223A-mutant 4T1 cells have been generated and characterized. Are there any cellular or molecular changes already observed in cell culture? Would an exclusive expression of Smurf1-T223A, in the absence of any WT form, affect cell proliferation and survival? Does the mutant form exert a dominant-negative effect? Would these experiments also work in other metastatic models of breast cancer, for example with MDA-MB-231 cells?

Reviewer #2 (Comments to the Authors (Required)):

In the manuscript, Jian et al uncovered a novel mechanism by which ERK and TGF β induce EMT. They provided evidence that active ERK can phosphorylate Smurf1 at Thr223. Smurf1 phosphorylation at Thr223 resulted in enhanced RhoA ubiquitination and degradation. Finally, they showed that Smurf1-T223A phosphorylation-deficient mutant significantly reduced TGF β induced EMT and lung metastasis.

Overall, the experiments were well executed with proper controls and the data are of high quality. While the study suggests a new mechanism for TGF β induced RhoA degradation, there are a number of key data needed to further strengthen the conclusions.

Key points:

1. As discussed in the introduction, degradation of a specific protein by an E3 ligase is usually regulated by phosphorylation of the substrate, such as p120, which led to recognition of p120 by Smurf1 for targeted degradation. Therefore, the proposed mechanism on Smurf1 phosphorylation to control RhoA degradation is interesting, yet unusual. Phosphorylation of Smurf1 on T223 would likely impact its interaction with many substrates beyond RhoA. To determine the specificity of Smurf1 T223 phosphorylation on RhoA degradation, analysis of other Smurf1 substrates, such as p120 is needed.
2. Another related point is that Erk activation occurs within 6hr upon TGF β treatment, while RhoA degradation becomes prominent only after 18-20hrs. Such slow RhoA degradation kinetics suggests that in addition to Erk activation, additional signal is needed to trigger RhoA degradation. In tumor cells expressing oncogenic Ras, it would be informative to ask whether RhoA degradation is triggered in the absence of TGF β .

Specific points:

1. Many blots in each figure need to be quantified to better report the levels of protein and phosphorylation.
2. Fig. 1 data are all based on overexpression and need endogenous protein confirmation for Erk/Smurf1 interaction.
3. Also Fig. 3A-3B suggest that endogenous Smurf1 is responsible for RhoA degradation in response to TGF β and Erk activation. Why is Fig. 3C and 3D show that overexpression of mutant Smurf1 in cells with endogenous Smurf1 led to reduced degradation? Is Smurf1 T223A dominant negative?
4. Fig. 3F is the key figure showing the importance of endogenous Smurf1 in mediating RhoA degradation. It is important to test whether Smurf1 knockdown blocks TGF β -induced RhoA degradation.
5. Although the evidence for Smurf1 phosphorylation resulted in enhanced RhoA degradation is strong. It is not sufficient to support "ERK-mediated Smurf1 phosphorylation is a prerequisite step for RhoA degradation". Several figures in the manuscript suggested exogenous Smurf1 expression can increase RhoA degradation without ERK phosphorylation. In Figure 3D and 4F, F/Smurf1 wt decreased RhoA degradation without TGF β treatment. Similarly, In Figure S4A, Myc/Smurf1 significantly increased RhoA ubiquitination and degradation without ERK2-R67S.
6. It has been shown that TGF β treatment can increase Smurf1 expression via MAPK-ERK signaling or Smad-SND11,2. Have the authors checked whether Smurf1 protein was upregulated with TGF β treatment? It would be good to know whether Smurf1 upregulation was involved in TGF β induced RhoA degradation.
7. In Figure 4E, the authors showed TGF β treatment led to polyubiquitination of RhoA, and this could also be attenuated by U0126 which is clearly shown by ub-RhoA staining. But why RhoA protein level didn't change with TGF β or U0126 treatment?

Reviewer #3 (Comments to the Authors (Required)):

The authors present an interesting study describing how ERK1/2 binds to HECT family E3 ligase Smurf1 but not Smurf2 both in vivo and in vitro and phosphorylates Smurf1 on Threonine 223 residue upon TGF-beta treatment. The authors further show that phosphorylated Smurf1 physically binds to RhoA, mediates its polyubiquitination and degradation, and finally promotes TGF-beta-dependent dissolution of tight-junctions during EMT, a process that can be blocked by ERK inhibition or point mutation of Threonine to Alanine on the 223 residue of Smurf1. Finally, the authors discover that overexpressing the T223A mutant form of Smurf1 in MDA-MB-231 cells decreases their migration ability in vitro, and the 4T1 cells with the T223A Smurf1 expression exhibit reduced metastatic potential in immunocompetent mice. The results presented in this manuscript identify a novel

interaction between ERK and Smurf1 and provide interesting observations about the ERK-Smurf1-RhoA pathway in TGF-beta-induced EMT. The study extends the current understanding of the function of ERK in EMT. However, some questions need to be addressed and some improvements should be done before the current manuscript being considered for publication.

Comments on main conclusions:

1. In the first section, by using co-ip and GST pull-down, the authors show that ERK1/2 interacts with Smurf1 in vivo and in vitro. The presented data are sufficient to support the conclusion.

Comments: the authors should specify whether the mass spec result was based on the indicated band in Figure 1A or all proteins from IP. It would also be helpful if the authors can include a list of proteins identified by mass spec.

2. In the second part, the authors show that TGF-beta treatment induces ERK-dependent phosphorylation of Smurf1. Further in vitro kinase assay and MS analysis demonstrate that ERK phosphorylates Smurf1 on Threonine 223 residue and point mutation of this residue to Alanine abolishes the phosphorylation. The data can strongly support the conclusion.

Comments: in Figure 2A, it looks like that the phosphorylation of Smurf1 reaches the maximum at 1hr post-treatment and decreases to basal level at 1.5hr, but the level of phospho-ERK remains the same. The authors should discuss the possible reasons for this reduction.

3. In the next two sections, the authors conduct a series of experiments to show the requirement of T223 phosphorylation of Smurf1 for its binding and ubiquitination of RhoA. The present data seem sufficient to support the conclusions. However, some questions need to be answered and additional experiments are suggested to strengthen the conclusion.

Question: previous studies (Wang et al., Regulation of cell polarity and protrusion formation by targeting RhoA for degradation, Science, 2003; Wang et al., Degradation of RhoA by Smurf1 ubiquitin ligase, Methods in Enzymology, 2006) have shown that Smurf1 can control the local RhoA level by targeting RhoA for degradation and regulate cell protrusion dynamic. These papers should be referenced and discussed in the manuscript. Further, Wang et al. Science 2003 showed that Smurf1 binds to nucleotide-free or GDP-bound RhoA and targets it for ubiquitination in vitro without being phosphorylated on Threonine 223. The authors should comment on this difference and explain why T223 phosphorylation is required for RhoA binding and degradation in their system.

Experiments: besides the biochemistry method, the authors should include immunostaining images showing the localization of Smurf1 and RhoA with or without TGF-beta treatment. Also, it would be interesting to show how ERK inhibition or expression of T223A Smurf1 influences the localization.

4. Lastly, by using immunostaining, migration, and invasion assays, the authors show that ERK-mediated T223 phosphorylation of Smurf1 is needed for the dissolution of cell tight junctions during EMT and the ability of cancer cells to invade. Further, the tumor allografts indicate that Smurf1 phosphorylation is critical for cancer metastasis. The data are strongly supportive, but some questions need to be answered to clarify the results.

Questions: in Figure 5A&B, did the authors observe any cell morphological changes in the bright field channel? In figure 5E, did the authors express human Smurf1 or murine Smurf1 in the 4T1 cells? In Figure S5B, no lung metastasis can be observed in both mice groups. Is it due to the different scales of bioluminescence intensity?

Additional comments:

5. The methods for some experiments are missing, such as the migration and invasion assay for MDA-MB-231 cells.

6. In the first paragraph of the introduction, full names of the growth factors should be specified.

7. The last sentence (TGF-beta signaling having.....) of the first paragraph and the second sentence (ERK MAPK could be) of the second paragraph of the introduction are too long and hard to understand. It is suggested to break these sentences into short ones.

8. In the last sentence (TGF-beta signaling having.....) of the first paragraph, "in regulation" should be "in the regulation of" and "all the scenario" should be "all the scenarios".

9. In the last section, "Smurf1-T223A expression breast cancer cells" should be "expressing".

Reviewer #1 (Comments to the Authors (Required)):

The critical function of MEK-ERK1/2 signaling in tumor initiation, progression and metastasis formation has been documented. Here, Zheng and co-workers have investigated the functional contribution of ERK1/2 signaling to EMT in MCF7 breast cancer cells and on tumor metastasis in the 4T1 syngeneic transplantation mouse model of breast cancer. Notably, they have set out to identify novel non-canonical pathways of ERK1/2 signaling. They report the identification of Smurf1 as a binding partner of ERK1/2 and that ERK1/2-mediated phosphorylation of Smurf1 at Thr223 promotes Smurf1-mediated poly-ubiquitination and degradation of RhoA, thus promoting EMT. Blockade of ERK1/2-mediated phosphorylation of Smurf1 and of RhoA degradation represses TGF β -induced EMT and experimental breast cancer metastasis.

The study reports novel insights into an ERK1/2 pathway, thus far neglected in EMT and metastasis research. The experimental approaches are thoughtfully designed, yet they are limited to biochemical and cellular analyses on almost exclusive MCF7 cells only. The results are carefully interpreted, yet may lack generality. Overall, this is an interesting and medically relevant report which seems still preliminary and in parts limited in its proof-of-concept experimentation. Also, a previous manuscript has reported a critical role of Smurf1 in EMT of ovarian cancer cells (Ref. 31). The authors should specifically discuss their results in the context of this recent publication.

Answer: Following the reviewer's suggestion, we have discussed their results in our manuscript.

Specific comments:

1. The selection of general reviews referred to in the Introduction is curious and rather not specific to the actual topic: For example, reference 23 deals with ERK signaling and depression....

Answer: We apologize for incorrectly citing some reviews. We reorganized our references and we have changed the text accordingly.

2. English Grammar and Style are not of high quality and the manuscript certainly needs adequate proofreading.

Answer: We appreciate the reviewer's suggestion, which are essential for the improvement of this manuscript. We carefully checked and revised our manuscript.

3. Figure 1: It seems important to show a co-immunoprecipitation of both endogenous proteins, ERK1/2 and Smurf1. In the data shown, one protein is always overexpressed and tagged.

Answer: As suggested by the reviewer, we examined the interaction between Smurf1 and ERK by co-immunoprecipitation, and ascertained that Smurf1 did bind to ERK endogenously (revised Figure 1C).

4. In the experiments shown in Figure 1, it is not clear whether the cells have been treated with

TGF β . If not, this would mean that ERK1/2 are already interacting with Smurf1 without activation of ERK1/2? Later in the manuscript in Figure 4A and B, it is shown that they interact only upon TGF β stimulation. What is then the mechanisms that allows the interaction only after TGF β stimulation, changes in localization, in ERK1/2 activity or in the recruitment of co-factors or else?

Answer: We apologize for the unclear description. Figure 1 showed that Smurf1 could bind to ERK without TGF β treatment. While, Figure 4A and B showed that TGF β treatment improved the interaction between Smurf1 and RhoA. Our data indicated that TGF β treatment could result in ERK activation followed by Smurf1 phosphorylation, thereby promoting the binding of Smurf1 to RhoA. Besides, phosphorylation of Smurf1 on Threonine 223 could lead to salt bridges formation between pThr223 of Smurf1 and His105 of RHOA (revised Figure S4A), thus, enhancing their interaction.

5. Figure S2E is shown before Figure S2C (order of appearance).

Answer: We apologize for the incorrect order of figures. We reorganized our figures and we have changed the text accordingly.

6. The authors argue that the loss of RhoA disrupts the Par6 polarity complexes and thus dissolves cortical actin. On the other hand, RhoA is a well-known player in the formation of stress fibers, which in turn are required for EMT and cell migration. How can the cells undergo EMT, when RhoA is lost?

Answer: The reviewer is correct that some evidence suggests that RhoA has a role in stress fibers formation and promoting EMT. While, RhoA is also essential for the maintenance of apical-basal polarity and cell-cell junction integrity (*Dev. Cell*, 2002, 3, 367; *Nature Cell Biol*, 2002, 4, 408), and it has been reported that TGF β -induced EMT requires a degradation of RhoA to disassemble cortical actin filaments (*Science*, 2005, 307, 1603-1609). Thus, RhoA might play different spatiotemporal roles in different context.

7. The number of repeats of each experiment should be mentioned in the figure legends. Some immunoblots are not of adequate quality to allow a quantitative interpretation (for example Figure 3D, F, H), better blots should be shown and actually quantified.

Answer: We apologize for the unclear description in our figure legend. Following the reviewer's suggestion, we revised the figure legend and repeated some experiments, and quantified the protein levels of RhoA and the phosphorylation levels of Smurf1 (revised Figure S2G, revised Figure 3D, revised Figure 3F, revised Figure 3J, revised Figure 3L, revised Figure S3A, revised Figure S3C, revised Figure S3F, revised Figure S3H).

8. The immunoblots shown in Figure 4E and F and S4A are not clear: Ubiquitinated proteins usually run as ladders above the non-ubiquitinated forms, hence the ladder of proteins immunoblotted for ubiquitin should go from bottom to top, not the other way around as shown here?

Answer: Smurf1 targets RhoA for polyubiquitination and degradation. Thus, the ubiquitinated

proteins usually have much high molecular weight, thereby, forming ladders mainly in the top. Besides, our data is consistent with previous studies (*Science*, 2003, 302, 1775)

9. The proteasomal degradation of RhoA, as claimed by the authors, should be demonstrated for example by using a proteasomal inhibitor or by time course and pulse chase experiments to assess RhoA protein stability.

Answer: According to the reviewer's suggestion, we examined whether proteasomal inhibitor MG132 could block ERK2-R67S-induced RhoA degradation, and we confirmed that MG132 notably attenuated RhoA degradation (revised Figure S3I). Besides, we conducted pulse chase experiment and ascertained that RhoA was much stable in Smurf1-T223A expressing cells (revised Figure S3G, Figure S3H).

10. Figure 5E and F: The bioluminescence images and their quantification are not clear: the primary tumors seem to have been covered in the bioluminescence imaging process, since their high activities would have impaired the analysis of the metastatic lesion. Still, without removal of the primary tumor, how can this be adequately quantified? There also seem to be metastatic lesions in the brain and less in the lung, the latter known to be the major target organ of 4T1 cells. It may be advised to remove the primary tumors when reaching a certain size and then assess metastasis formation. Along these lines, it may be important to learn whether the Smurf-1 phosphorylation-deficient tumors show any difference in histopathological parameters, such as invasion, proliferation, apoptosis etc.

Furthermore, it is not clear how the Smurf1-WT and T223A-mutant 4T1 cells have been generated and characterized. Are there any cellular or molecular changes already observed in cell culture? Would an exclusive expression of Smurf1-T223A, in the absence of any WT form, affect cell proliferation and survival? Does the mutant form exert a dominant-negative effect? Would these experiments also work in other metastatic models of breast cancer, for example with MDA-MB-231 cells?

Answer: Following the reviewer's suggestion, we conducted in vivo metastasis assay in mice and the primary tumors were resected when they reached a volume greater than 300 mm³. By using HE staining and counting the numbers of lung metastasis nodules, we confirmed that Smurf1-T223A mutant significantly blocked lung metastasis (revised Figure 5E-5F). Besides, we further tested whether Smurf1-T223A mutant affected cell proliferation or apoptosis of primary tumors by IHC, and we found that Smurf1-T223A mutant did not affect cell proliferation or apoptosis (revised Figure S5C).

We apologize for the unclear description. Actually, we did depict how Smurf1-WT and T223A-mutant 4T1 cells are generated in the figure legend. We now describe this experiment procedure in the main text for the convenience of readers. As suggested by the reviewer, we examined the key cell junction markers and checked the cell morphology of Smurf1-WT and T223A-mutant 4T1 cells by Immunofluorescence assay, and ascertained that T223A-mutant didn't affect cell junction formation or cell morphology as compared to Smurf1-WT (Fig. for reviewers A). Then we tested whether T223A-mutant influenced cell proliferation by using CCK8 assay, and we found that T223A-mutant didn't have any effect on cell proliferation (Fig. for reviewers B). According to the reviewer's suggestion, we further repeated in vivo metastasis assay using

MDA-MB-231 cells, and confirmed that T223A-mutant also significantly attenuated tumor metastasis in that model (Fig. for reviewers C).

Reviewer #2 (Comments to the Authors (Required)):

In the manuscript, Jian et al uncovered a novel mechanism by which ERK and TGF β induce EMT. They provided evidence that active ERK can phosphorylate Smurf1 at Thr223. Smurf1 phosphorylation at Thr223 resulted in enhanced RhoA ubiquitination and degradation. Finally, they showed that Smurf1-T223A phosphorylation-deficient mutant significantly reduced TGF β induced EMT and lung metastasis.

Overall, the experiments were well executed with proper controls and the data are of high quality. While the study suggests a new mechanism for TGF β induced RhoA degradation, there are a number of key data needed to further strengthen the conclusions.

Key points:

1. As discussed in the introduction, degradation of a specific protein by an E3 ligase is usually regulated by phosphorylation of the substrate, such as p120, which led to recognition of p120 by Smurf1 for targeted degradation. Therefore, the proposed mechanism on Smurf1 phosphorylation to control RhoA degradation is interesting, yet unusual. Phosphorylation of Smurf1 on T223 would likely impact its interaction with many substrates beyond RhoA. To determine the specificity of Smurf1 T223 phosphorylation on RhoA degradation, analysis of other Smurf1 substrates, such as p120 is needed.

Answer: As suggested by the reviewer, we examined whether phosphorylation of Smurf1 on T223 influenced the degradation of RhoB, another substrate of Smurf1. We found that Smurf1 phosphorylation didn't affect RhoB turnover (Fig. for reviewers D). Besides, previous study showed that Smurf1 targets p120-catenine for monoubiquitination and cytosol translocation not degradation (*science advances*, 2020, 6, 4). Thus, we tested whether Smurf1 phosphorylation affected p120-catenine monoubiquitination, and discovered that Smurf1-T223A mutant and Smurf1-T223D mutant didn't influence p120-catenine monoubiquitination as compared to Smurf1-WT (Fig. for reviewers E).

2. Another related point is that Erk activation occurs within 6hr upon TGF β treatment, while RhoA degradation becomes prominent only after 18-20hrs. Such slow RhoA degradation kinetics suggests that in addition to Erk activation, additional signal is needed to trigger RhoA degradation. In tumor cells expressing oncogenic Ras, it would be informative to ask whether RhoA degradation is triggered in the absence of TGF β .

Answer: According to the reviewer's suggestion, we examined whether Ras overexpression could influence steady-state levels of RhoA. Interestingly, Ras overexpression also led to RhoA degradation (Fig. for reviewers F). This is conceivable, since Ras overexpression would result in activation of ERK and Smurf1 phosphorylation.

Specific points:

1. Many blots in each figure need to be quantified to better report the levels of protein and phosphorylation.

Answer: Thanks for the reviewer's suggestion. As described above, we repeated some experiments, and quantified the protein levels of RhoA and the phosphorylation levels of Smurf1 (revised Figure S2G, revised Figure 3D, revised Figure 3F, revised Figure 3J, revised Figure 3L, revised Figure S3A, revised Figure S3C, revised Figure S3F, revised Figure S3H).

2. Fig. 1 data are all based on overexpression and need endogenous protein confirmation for Erk/Smurf1 interaction.

Answer: As suggested by the reviewer, we examined the interaction between Smurf1 and ERK by co-immunoprecipitation, and ascertained that Smurf1 did interact with ERK endogenously (revised Figure 1C).

3. Also Fig. 3A-3B suggest that endogenous Smurf1 is responsible for RhoA degradation in response to TGF β and Erk activation. Why is Fig. 3C and 3D show that overexpression of mutant Smurf1 in cells with endogenous Smurf1 led to reduced degradation? Is Smurf1 T223A dominant negative?

Answer: According to previous studies (*Science*, 2005, 307, 1603-1609; *science advances*, 2020, 6, 4) and our data in this research, TGF β treatment recruits Smurf1 to cell junction regions and targets localized RhoA for degradation. Thus, Overexpression of Smurf1-T223A mutant in cells would reduce the amount of Smurf1 that could interact with RhoA after TGF β treatment, leading to reduced degradation of RhoA. To some extent, we think Smurf1-T223A mutant is a dominant negative form.

4. Fig. 3F is the key figure showing the importance of endogenous Smurf1 in mediating RhoA degradation. It is important to test whether Smurf1 knockdown blocks TGF β -induced RhoA degradation.

Answer: As suggested by the reviewer, we examined RhoA turnover after TGF β treatment in Smurf1-knockdown MCF-7 cells. As predicted, Smurf1 knockdown markedly blocked TGF β -induced RhoA degradation (revised Figure S3E, Figure S3F).

5. Although the evidence for Smurf phosphorylation resulted in enhanced RhoA degradation is strong. It is not sufficient to support "ERK-mediated Smurf1 phosphorylation is a prerequisite step for RhoA degradation". Several figures in the manuscript suggested exogenous Smurf1 expression can increase RhoA degradation without ERK phosphorylation. In Figure 3D and 4F, F/Smurf1 wt decreased RhoA degradation without TGF β treatment. Similarly, In Figure S4A, Myc/Smurf1 significantly increased RhoA ubiquitination and degradation without ERK2-R67S.

Answer: Our research demonstrated that ERK-mediated Smurf1 phosphorylation could promote

the interaction between Smurf1 and RhoA, leading to RhoA polyubiquitination and degradation. Overexpressing Smurf1 and RhoA in cells, as described in previous Figure 3D, 4F and previous Figure S4A, could lead to sufficient amount of Smurf1 binding to RhoA, resulting in RhoA polyubiquitination and degradation.

6. It has been shown that TGF β treatment can increase Smurf1 expression via MAPK-ERK signaling or Smad-SND11,2. Have the authors checked whether Smurf1 protein was upregulated with TGF β treatment? It would be good to know whether Smurf1 upregulation was involved in TGF β induced RhoA degradation.

Answer: Following the reviewer's suggestion, we checked Smurf1 protein levels after TGF β treatment and found that Smurf1 levels were increased 18hrs after TGF β treatment (Fig. for reviewers G). Thus, TGF β treatment could trigger both Smurf1 phosphorylation and upregulation, leading to RhoA degradation.

7. In Figure 4E, the authors showed TGF β treatment led to polyubiquitination of RhoA, and this could also be attenuated by U0126 which is clearly shown by ub-RhoA staining. But why RhoA protein level didn't change with TGF β or U0126 treatment?

Answer: The reviewer is correct that RhoA protein levels was down-regulated after ubiquitination in response to TGF β treatment and up-regulated after RhoA ubiquitination was blocked by U0126. We apologize for the incorrect data and repeated this experiment (revised Figure 4E).

Reviewer #3 (Comments to the Authors (Required)):

The authors present an interesting study describing how ERK1/2 binds to HECT family E3 ligase Smurf1 but not Smurf2 both in vivo and in vitro and phosphorylates Smurf1 on Threonine 223 residue upon TGF-beta treatment. The authors further show that phosphorylated Smurf1 physically binds to RhoA, mediates its polyubiquitination and degradation, and finally promotes TGF-beta-dependent dissolution of tight-junctions during EMT, a process that can be blocked by ERK inhibition or point mutation of Threonine to Alanine on the 223 residue of Smurf1. Finally, the authors discover that overexpressing the T223A mutant form of Smurf1 in MDA-MB-231 cells decreases their migration ability in vitro, and the 4T1 cells with the T223A Smurf1 expression exhibit reduced metastatic potential in immunocompetent mice. The results presented in this manuscript identify a novel interaction between ERK and Smurf1 and provide interesting observations about the ERK-Smurf1-RhoA pathway in TGF-beta-induced EMT. The study extends the current understanding of the function of ERK in EMT. However, some questions need to be addressed and some improvements should be done before the current manuscript being considered for publication.

Comments on main conclusions:

1. In the first section, by using co-ip and GST pull-down, the authors show that ERK1/2 interacts

with Smurf1 in vivo and in vitro. The presented data are sufficient to support the conclusion.

Comments: the authors should specify whether the mass spec result was based on the indicated band in Figure 1A or all proteins from IP. It would also be helpful if the authors can include a list of proteins identified by mass spec.

Answer: We only sent the indicated band in Figure 1A for mass spectrometry analysis. Following the reviewer's suggestion, we will change the figure legend of Figure 1A accordingly and upload a list of proteins identified by mass spec.

2. In the second part, the authors show that TGF-beta treatment induces ERK-dependent phosphorylation of Smurf1. Further in vitro kinase assay and MS analysis demonstrate that ERK phosphorylates Smurf1 on Threonine 223 residue and point mutation of this residue to Alanine abolishes the phosphorylation. The data can strongly support the conclusion.

Comments: in Figure 2A, it looks like that the phosphorylation of Smurf1 reaches the maximum at 1hr post-treatment and decreases to basal level at 1.5hr, but the level of phospho-ERK remains the same. The authors should discuss the possible reasons for this reduction.

Answer: Phosphorylation of Smurf1 reaches the maximum at 1hr post-treatment and then quickly decreases to basal level. The reason, we think, is that a yet unknown phosphatase would be recruited and dephosphorylate the phosphorylated Smurf1, preventing the ERK-smurf1 signaling from over activation. Thus, in our opinion, phosphorylation of Smurf1 is dynamically regulated.

3. In the next two sections, the authors conduct a series of experiments to show the requirement of T223 phosphorylation of Smurf1 for its binding and ubiquitination of RhoA. The present data seem sufficient to support the conclusions. However, some questions need to be answered and additional experiments are suggested to strengthen the conclusion.

Question: previous studies (Wang et al., Regulation of cell polarity and protrusion formation by targeting RhoA for degradation, Science, 2003; Wang et al., Degradation of RhoA by Smurf1 ubiquitin ligase, Methods in Enzymology, 2006) have shown that Smurf1 can control the local RhoA level by targeting RhoA for degradation and regulate cell protrusion dynamic. These papers should be referenced and discussed in the manuscript. Further, Wang et al. Science 2003 showed that Smurf1 binds to nucleotide-free or GDP-bound RhoA and targets it for ubiquitination in vitro without being phosphorylated on Threonine 223. The authors should comment on this difference and explain why T223 phosphorylation is required for RhoA binding and degradation in their system.

Answer: Following the reviewer's suggestion, we cited and discussed the papers above in our manuscript. Wang et al. Science 2003 showed that Smurf1 binds to nucleotide-free or GDP-bound RhoA, They transfected F/Smurf1 and HA/RhoA in HEK293 cells or purified the proteins in vitro and examined the interaction between F/Smurf1 and HA/RhoA. However, in this study, we conducted the experiment using MCF-7 cells. MCF-7 cells are epithelial cells with apical-basal

polarity. Actually, we also found that Smurf1 could slightly bind to RhoA N19 in MCF-7 cells (Fig. for reviewers H). Besides, in epithelial cells, Smurf1 could target the localized RhoA for polyubiquitination and degradation only after it has been recruited to cell junction region in response to TGF β treatment (*Science*, 2005, 307, 1603-1609). Thus, our data is not conflict with previous studies.

Experiments: besides the biochemistry method, the authors should include immunostaining images showing the localization of Smurf1 and RhoA with or without TGF-beta treatment. Also, it would be interesting to show how EKR inhibition or expression of T223A Smurf1 influences the localization.

Answer: As suggested by the reviewer, we have tried to perform the Immunofluorescence assay to show the localization of Smurf1 and RhoA but we failed to get the positive data since our endogenous RhoA antibody is not good enough for this experiment. Besides, Smurf1 could only target the junction localized nucleotide-free or GDP-bound RhoA, which is really a very small part as compared to the total amount of RhoA, for polyubiquitination and degradation(*Science*, 2003, 302, 1775; *Science*, 2005, 307, 1603-1609). Instead, we carried out protein-protein Molecular Docking Simulation, and found that phosphorylation of Smurf1 on Threonine 223 could lead to salt bridges formation between pThr223 of Smurf1 and His105 of RhoA (revised Figure S4A), thus, enhancing their interaction.

4. Lastly, by using immunostaining, migration, and invasion assays, the authors show that ERK-mediated T223 phosphorylation of Smurf1 is needed for the dissolution of cell tight junctions during EMT and the ability of cancer cells to invade. Further, the tumor allografts indicate that Smurf1 phosphorylation is critical for cancer metastasis. The data are strongly supportive, but some questions need to be answered to clarify the results.

Questions: in Figure 5A&B, did the authors observe any cell morphological changes in the bright field channel? In figure 5E, did the authors express human Smurf1 or murine Smurf1 in the 4T1 cells? In Figure S5B, no lung metastasis can be observed in both mice groups. Is it due to the different scales of bioluminescence intensity?

Answer: We did not observe any cell morphological changes between Smurf1-WT and Smurf1-T223A mutant expressing cells in the bright field channel (Fig. for reviewers I). In previous figure 5E, we expressed human Smurf1. In Figure S5B, no lung metastasis was observed due to different bioluminescence intensities. The bioluminescence intensity of primary tumors was much higher than the lung metastasis.

Additional comments:

5. The methods for some experiments are missing, such as the migration and invasion assay for MDA-MB-231 cells.

Answer: Answer: We apologize for the unclear description. We will add these methods in our

manuscript.

6. In the first paragraph of the introduction, full names of the growth factors should be specified.

Answer: According to the reviewer's suggestion, we have changed the text accordingly.

7. The last sentence (TGF-beta signaling having.....) of the first paragraph and the second sentence (ERK MAPK could be) of the second paragraph of the introduction are too long and hard to understand. It is suggested to break these sentences into short ones.

Answer: According to the reviewer's suggestion, we have changed the text accordingly.

8. In the last sentence (TGF-beta signaling having.....) of the first paragraph, "in regulation" should be "in the regulation of" and "all the scenario" should be "all the scenarios".

Answer: According to the reviewer's suggestion, we have changed the text accordingly.

9. In the last section, "Smurf1-T223A expression breast cancer cells" should be "expressing".

Answer: According to the reviewer's suggestion, we have changed the text accordingly.

Figure for reviewers. (A, B) Smurf1-knockdown 4T1 cells transfected with FLAG/Smurf1-WT, or FLAG/Smurf1-T223A were subjected to immunofluorescence (IF) assay to detect cell-cell junctions. Scale bar, 10 μ m (A). CCK8 assay was used to determine cell proliferation (B). (C) Smurf1-knockdown MDA-MB-231 cells transfected with FLAG/Smurf1-WT, or FLAG/Smurf1-T223A (1×10^6) were

orthotopically injected into the mammary fat pad of female BALB/c mice (n=6 mice per group). For metastasis assays, tumors were surgically resected when they reached a volume greater than 300 mm³, 25 days after injection the mice were sacrificed. Lung sections were stained by Hematoxylin and eosin (H&E), and the numbers of lung metastasis nodules were counted and presented as mean ± SD of six mice per group.

(D) HEK293T cells transfected with HA/RhoB and FLAG/Smurf1-WT, FLAG/Smurf1-T223A or FLAG/Smurf1-T223D mutant were subjected to IB to detect HA/RhoB protein levels. (E) HEK293T cells transfected with FLAG/p120-catenine, HA/Ub and Myc/Smurf1-WT, Myc/Smurf1-T223A or Myc/Smurf1-T223D were subjected to ubiquitination assay to detect FLAG/p120-catenine ubiquitination. (F) HEK293T cells transfected with HA/Kras-WT or HA/Kras-V12 mutant were subjected to IB to detect the protein levels of RhoA. (G) MCF-7 cells were treated with or without TGFβ for the indicated time and subjected to IB to detect Smurf1 protein levels. (H) MCF-7 cells transfected with HA/Smurf1 and Flag/RhoA N19 were subjected to anti-Flag IP and IB to detect the associated HA/Smurf1. (I) Smurf1-knockdown MCF-7 cells transfected with FLAG/Smurf1-WT, or FLAG/Smurf1-T223A were visualized by phase contrast microscopy. Scale bar, 50 μm.

May 9, 2022

RE: Life Science Alliance Manuscript #LSA-2021-01330-TR

Dr. Qingang Wu
First Affiliated Hospital Zhejiang University
Zhejiang Provincial Key Laboratory of Pancreatic Disease, The First Affiliated Hospital, and Institute of Translational Medicine,
Zhejiang University School of Medicine
Hangzhou 310029, China

Dear Dr. Wu,

Thank you for submitting your revised manuscript entitled "ERK-Smurf1-RhoA signaling is critical for TGF-beta-driven EMT and tumor metastasis". We would be happy to publish your paper in Life Science Alliance pending final revisions necessary to meet our formatting guidelines.

- please address Reviewer 3's final comments
- please add the Twitter handle of your host institute/organization as well as your own or/and one of the authors in our system
- please use the [10 author names, et al.] format in your references (i.e. limit the author names to the first 10)

Figures:

- please include molecular weights next to all blots
- Figure 1A: is there a splice between the columns?
- Figure 5E needs a scale bar

A. FINAL FILES:

B. MANUSCRIPT ORGANIZATION AND FORMATTING:

Sincerely,

Reviewer #1 (Comments to the Authors (Required)):

The authors have adequately responded to the reviewers' comments. Notably, they have performed a substantial amount of additional experiments, including a large number of detailed biochemical analyses and in vivo tumor transplantation experiments. Together, the results now sufficiently support the conclusions drawn by the authors, and the manuscript provides important new insights which are of interest to a wide audience.

Reviewer #2 (Comments to the Authors (Required)):

The authors have addressed all issues raised previously and the manuscript is very well revised and ready for publication.

Reviewer #3 (Comments to the Authors (Required)):

The authors addressed most of the reviewers' comments and strengthened the manuscript. However, some remaining questions need to be answered before publication.

Specific comments:

1. The authors repeated some experiments and provided quantification of the protein levels. However, no statistical analysis was performed to demonstrate the significance. It is suggested to add statistics to these plots (Fig.3D, 3F, 3J, 3L, S3C, S3F, and S3H).

2. The results shown in Figure 3E&F do not fully match that of Figure 3C&D. In figure 3C&D, the authors suggested that exogenous expression of T223A smurf1 has a dominant-negative effect and rescues RhoA from degradation to some extent. However, such an effect is not obvious in Figure 3E&F, in which the relative RhoA level is close to 40% for both WT and T223A. The authors should explain this difference.

3. The authors repeated the metastasis experiment with MDA-MB-231 cells expressing WT-Smurf1 or T223A-Smurf1 and demonstrated less metastasis formed by cells with mutant Smurf1. This result is important and should be included in Figure S5. In the figure caption, the authors state that the MDA-MB-231 were orthotopically injected into female Balb/c mice. Are these Balb/c nude mice or immunocompetent?

-please address Reviewer 3's final comments

Yes

-please add the Twitter handle of your host institute/organization as well as your own or/and one of the authors in our system

Sorry, Twitter is not available in our country.

-please use the [10 author names, et al.] format in your references (i.e. limit the author names to the first 10)

According to the suggestion, we have changed the text accordingly.

Figures:

-please include molecular weights next to all blots

According to the suggestion, we have added the molecular weights next to all blots.

-Figure 1A: is there a splice between the columns?

No, there is no splice between the columns in Figure 1A

-Figure 5E needs a scale bar

According to the suggestion, we have added a scale bar in Figure 5E.

Reviewer #3 (Comments to the Authors (Required)):

The authors addressed most of the reviewers' comments and strengthened the manuscript. However, some remaining questions need to be answered before publication.

Specific comments:

1. The authors repeated some experiments and provided quantification of the protein levels. However, no statistical analysis was performed to demonstrate the significance. It is suggested to add statistics to these plots (Fig.3D, 3F, 3J, 3L, S3C, S3F, and S3H).

Answer: According to the reviewer's suggestion, we have shown the statistical significance in the figures accordingly.

2. The results shown in Figure 3E&F do not fully match that of Figure 3C&D. In figure 3C&D, the authors suggested that exogenous expression of T223A smurf1 has a dominant-negative effect and rescues RhoA from degradation to some extent. However, such an effect is not obvious in Figure 3E&F, in which the relative RhoA level is close to 40% for both WT and T223A. The authors should explain this difference.

Answer: According to previous studies (*Science*, 2005, 307, 1603-1609; *science advances*, 2020, 6, 4) and our data in this research, TGF β treatment recruits Smurf1 to cell junction regions and targets localized RhoA for degradation. Thus, Overexpression of Smurf1-T223A mutant in cells would reduce the amount of Smurf1 that could interact with RhoA after TGF β treatment, leading to reduced degradation of RhoA (Figure 3C and 3D). In Figure 3E and 3F, we demonstrated that Smurf1-T223D could further decrease the protein levels of RhoA than that of Smurf1-WT and Smurf1-T223A. This data doesn't contradict with the data in Figure 3C and 3D. In Figure 3C,

without TGF β treatment the protein levels of RhoA is almost the same in Smurf1-WT expressing cells and Smurf1-T223A expressing cells.

3. The authors repeated the metastasis experiment with MDA-MB-231 cells expressing WT-Smurf1 or T223A-Smurf1 and demonstrated less metastasis formed by cells with mutant Smurf1. This result is important and should be included in Figure S5. In the figure caption, the authors state that the MDA-MB-231 were orthotopically injected into female Balb/c mice. Are these Balb/c nude mice or immunocompetent?

Answer: According to the reviewer's suggestion, we have changed the text accordingly. The mice we used are Balb/c nude mice.

May 17, 2022

RE: Life Science Alliance Manuscript #LSA-2021-01330-TRR

Dr. Qingang Wu
First Affiliated Hospital Zhejiang University
Zhejiang Provincial Key Laboratory of Pancreatic Disease, The First Affiliated Hospital, and Institute of Translational Medicine,
Zhejiang University School of Medicine, Hangzhou 310029, China

Dear Dr. Wu,

Thank you for submitting your Research Article entitled "ERK-Smurf1-RhoA signaling is critical for TGF-beta-driven EMT and tumor metastasis". It is a pleasure to let you know that your manuscript is now accepted for publication in Life Science Alliance. Congratulations on this interesting work.

DISTRIBUTION OF MATERIALS:

Again, congratulations on a very nice paper. I hope you found the review process to be constructive and are pleased with how the manuscript was handled editorially. We look forward to future exciting submissions from your lab.

Sincerely,
